Species and size diversity in protective services offered by coral guard-crabs

McKeon C. Seabird 1 2 mckeons@si.edu
Moore Jenna M. 1
1 Florida Museum of Natural History, University of Florida , Gainesville, FL , USA
2 Smithsonian Institution Marine Science Network, Smithsonian Marine Station , Fort Pierce, FL , USA
Ford Alex
Electronic publication date: 2014 Sep 30
Publication date: 2014
Volume: 2
Electronic Location ID: e574
Received 2014 Feb 20; Accepted 2014 Aug 22
Copyright: © 2014 McKeon and Moore
Copyright year: 2014
Copyright holder: McKeon and Moore
License: This is an open access article distributed under the terms of the Creative Commons Attribution License, which permits unrestricted use, distribution, and reproduction in any medium, provided the original author and source are credited.
License URL: https://creativecommons.org/licenses/by/3.0/

Keywords: Functional diversity, Functional equivalence, Associational refuge, Complementarity, Mutualism, Symbiosis, Trapezia, Partner benefits, Acanthaster, Pocillopora

Funding: University of Florida BIOCODE Mo’orea Project This project was funded by an Alumni Fellowship from University of Florida to CS McKeon, and the BIOCODE Mo’orea project to G Paulay. Logistical support was provided by the BIOCODE Mo’orea Project. The funders had no role in study design, data collection and analysis, decision to publish, or preparation of the manuscript.

==============================
Coral guard-crabs in the genus Trapezia are well-documented defenders of their pocilloporid coral hosts against coral predators such as the Crown-of-Thorns seastar (Acanthaster planci complex). The objectives of this study were to examine the protective services of six species of Trapezia against corallivory, and the extent of functional diversity among these Trapezia species.

Studies conducted in Mo’orea, French Polynesia showed the Trapezia—coral mutualism protected the host corals from multiple predators through functional diversity in the assemblage of crab symbionts. Species differed in their defensive efficacy, but species within similar size classes shared similar abilities. Smaller-size Trapezia species, which were previously thought to be ineffective guards, play important defensive roles against small corallivores.

We also measured the benefits of this mutualism to corals in the midst of an Acanthaster outbreak that reduced the live coral cover on the fore reef to less than 4%. The mutualism may positively affect the reef coral demography and potential for recovery during adverse predation events through shelter of multiple species of small corals near the host coral. Our results show that while functional diversity is supported within the genus, some Trapezia species may be functionally equivalent within the same size class, decreasing the threat of gaps in coral protection caused by absence or replacement of any single Trapezia species.

Introduction

Mutualistic symbioses stabilize and increase biodiversity in an ecosystem by: (1) reducing antagonism between host and symbiont, eliminating “arms races” that could result in extinction; (2) increasing the resistance of each species to stressors external to the relationship; and (3) enhancing survival benefits for many species in complementary assemblages of mutualists and mutualistic networks. Mutualisms often occur with more than one species providing benefits to a host species, and community diversity within symbiotic suites has been largely overlooked (Howe, 1984). High mutualist diversity may confer greater survival benefits to the host species than a single mutualist (Stachowicz & Whitlatch, 2005; Baskett, Gaines & Nisbet, 2009), especially if functional diversity exists in the system. Whole communities may have net beneficial effects for a shared host that can be considered mutualistic in nature (Bracken, Gonzalez-Dorantes & Stachowicz, 2007; Stachowicz et al., 2008). Studies on the interactions between mutualist species are underrepresented in the literature (Morris et al., 2007), but relevant and central to understanding diverse systems (McKeon et al., 2012). Mutualist diversity within symbiotic suites drives variation in partner benefits, strongly impacting the extent and strength of these ecological partnerships (Schemske & Horvitz, 1984; Addicott, 1986; Meunier et al., 1999; Correa & Baker, 2009).

Functional diversity and equivalence in ecological roles of similar species have become focal points in our understanding of community structure and the generation and maintenance of biodiversity (Harris, 1995; Loreau, Naeem & Inchausti, 2002; Loreau, 2004). Diversity has been simplified into ecological units of ‘functional groups’, but studies testing the equivalence of species within functional groups have shown mixed results (Chalcraft & Resetarits, 2003). The few studies to date comparing ecological function of closely related species under the same environmental conditions have supported the idea that functional diversity, rather than equivalence, may be the norm (Resetarits & Chalcraft, 2007). Despite this evidence, the assumptions of functional equivalence underlying the construction of functional groups have remained a central premise in much of theoretical ecology, from Neutral Theory (Hubbell, 2005) to Food Web Theory (Menge & Sutherland, 1987; Leibold & McPeek, 2006).

Testing for functional diversity and equivalence is most tractable and applicable in clades of closely related or morphologically similar species (Leibold & McPeek, 2006). Crabs of the genus Trapezia differ in size but exhibit few other morphological differences. Trapezia are defensive mutualists of coral hosts in the family Pocilloporidae (Pearson & Endean, 1969; Glynn, 1976; Glynn, 1987). These crabs have been implicated in providing a number of ecological services to their host corals: repulsion of corallivores (Glynn, 1987; Pratchett, 2001), removal of sediment (Stewart et al., 2006), and alleviation of the impacts of vermetid snails (Stier et al., 2010). The functional differences between different species of this well-known marine mutualist genus have not yet been examined in detail.

Trapezia are part of an assemblage of specialized associates of pocilloporid corals. Pocillopora make up a large percentage of reef cover in lagoonal and fore reef systems in the Indo-Pacific. Fourteen of the 22 described Trapezia species occur in French Polynesia (Castro, 1997). Symbiont community composition varies with coral species, growth form, reef zone, and community membership (Odinetz, 1983). The maximum size of Trapezia is constrained by host interbranch width (Adams, Edwards & Emberton, 1985; Huber & Coles, 1985). A single mating pair of a given Trapezia species typically occupies each coral colony, although each colony can host multiple species and additional juveniles. Larger coral colonies may host pairs of up to five species of Trapezia, while smaller coral colonies may only shelter a single species. The benefits to the Trapezia include shelter and nutrition in the form of lipids sequestered in the tips of the polyp tentacles, which the crabs graze (Stimson, 1990).

The objectives of this study were: (1) to evaluate the services of multiple Trapezia species to their host corals; (2) to make interspecific comparisons of functional roles in Trapezia; and (3) to assess the effect of symbiont size on functional role. We compared the defensive efficacy of several species and size classes of Trapezia against two corallivores in lab experiments. We also examined the role of the Pocillopora–Trapezia mutualism in mitigating the effects of Acanthaster planci, the Crown-of-Thorns sea star, during a natural outbreak event.

Methods

Study locations

Corallivore defense studies were conducted in Mo’orea, French Polynesia, at the Richard B. Gump Research Station of the University of California, utilizing the flow-through seawater system and a fore reef field site.

Study organisms

Pocillopora species make up a large percentage of reef cover in lagoonal and fore reef systems in the Indo-Pacific. The bulk of this is composed of coral colonies with morphologies in three coarse groupings. The Pocillopora verrucosa—P. meandrina group, referred to in this manuscript as “Pocillopora verrucosa”, are mid-sized pocilloporids, rarely exceeding 15 cm in colony height and typically occupied by small- to medium-sized Trapezia species. A second morphological group, the P. eydouxi—P. woodjonesi group (hereafter “Pocillopora eydouxi”), is larger in stature, commonly exceeding 60 cm in colony height, and has substantially broader interbranch widths. These corals host the entire size range of Trapezia species; from the smallest at branch junctures and the colony’s base, to the largest, which actively transit the openings between branches. The third morphological grouping is composed of species currently assigned to Pocillopora damicornis, a finely branched morphotype that exhibits extreme environmental variation across reef microhabitats (Veron & Pichon, 1976).

Species-level distinctions across the geographic range of Pocillopora remain unresolved at morphological, genetic, and taxonomic levels (Veron & Pichon, 1976; Veron, 2000; Combosch et al., 2008). As such, we chose to use internally consistent morphological groupings in the experiments, but do not have further identification of the entities involved.

We investigated Trapezia defense against three corallivores: the Crown-of-Thorns seastar Acanthaster “planci” (Linnaeus 1758), a species complex (Vogler et al., 2008); the seastar Culcita novaeguineae Müller & Troschel 1842; and the muricid gastropod Drupella cornus (Röding 1798). For simplicity, we will refer to these species by their generic name only hereafter.

These species are the most common coral predators on Indo-Pacific reefs. Acanthaster often occurs in high densities during population booms (Birkeland, 1989). Acanthaster is capable of consuming all reef corals, but does exhibit feeding preferences (Pratchett, 2007; Pratchett et al., 2009). Culcita is a generalist predator of sessile organisms, including corals (Glynn & Krupp, 1986). Drupella is a specialized corallivore whose impact on reef corals is second only to Acanthaster (Turner, 1994). The three species feed on corals nocturnally, and leave visible ‘scars’ of exposed coral skeleton in the course of a single feeding event.

Experiments on defense utilized single reproductive pairs of Trapezia punctimanus Odinetz 1984 (a species of generally smaller size within the study area); T. bidentata (Forskål, 1775) and T. serenei Odinetz 1984 (medium sized species); and T. flavopunctata Eydoux & Souleyet, 1842 (a large species). Small carapace-width (CW) pairs of T. serenei were also used as a ‘small’ species in the experiments with Drupella, as crabs of this species reach sexual maturity at much smaller sizes when living within small coral colonies. The first author identified the crabs to species, relying on Castro, Ng & Ahyong (2004), the original species descriptions, and molecular data (McKeon, 2010) from voucher specimens deposited at the Florida Museum of Natural History.

Experimental design

Four species of Trapezia were used in experiments evaluating defense against the three corallivores. These were: T. serenei Odinetz 1984 (two size classes: small 4–6 mm CW, and medium 9–11 mm CW), T. punctimanus Odinetz 1984 (small size class: 4–6 mm CW), T. bidentata (Forskål 1775); (medium size class: 9–11 mm CW), and T. flavopunctata Eydoux & Souleyet 1842 (large size class: >11 mm CW). The differences in size between the three species of corallivore (Acanthaster planci (a large predatory asteroid), Culcita novaeguineae (a medium sized asteroid predator), and Drupella cornus (a small predatory snail)) allowed for evaluation of the impact of size of both predator and defender in the defensive efficacy of species of Trapezia. The extreme difference in size among corallivore, coral host, and crabs necessitated different experimental approaches to assess the range of defensive behaviors. Large individuals of Acanthaster in particular were unreliable predators in lab feeding trials, but were easy to track during field experiments, while Culcita and Drupella would readily attempt feeding during a single night in the lab chambers, but were difficult to trace in the field.

During 2008 and 2009, Mo’orea experienced an outbreak of Acanthaster, allowing field manipulations to test the effect of Trapezia defense on Acanthaster predation. By September 2008, the majority of live coral had been eaten, with the notable exception of Pocillopora eydouxi Milne Edwards & Haime 1860, which occurs on the forereef and hosts pairs of T. flavopunctata as well as a suite of other symbiotic species. A closely related species, Pocillopora damicornis Linnaeus, 1758 is a highly favored food in the diet of Acanthaster (Pratchett, 2007). Smaller Pocillopora verrucosa (Ellis & Solander 1786) had been consumed in near totality, and least-favored taxa such as Porites and soft corals were also being eaten.

Drupella cornus

Two sets of experiments tested the defensive efficacy of two size classes of Trapezia serenei (small adult crabs 4–6 mm CW, and medium adult crabs 9–10 mm CW) and one size class of T. punctimanus (small—4–6 mm CW) against the corallivorous snail Drupella cornus. Two 135 L aquaria equipped with flowing seawater served as experimental chambers. Pocillopora verrucosa colonies were gathered from the back reef environment on the day of the experiments. Small coral colonies had a mean volume of 260 ± 136 cm3, and the larger coral colonies had a mean volume of 3,706 ± 1,057 cm3. Exosymbionts and other animals were removed from the corals using wooden skewers.

Experimental treatments for the small size class (4–6 mm CW) of crabs were: all symbionts removed (n = 22), small T. serenei (n = 22), and T. punctimanus (n = 22). Experimental treatments for the medium size class (9–10 mm CW) only included T. serenei and a removal treatment, because T. punctimanus does not occur in an equivalent size range. Both experiments were conducted as described below.

Forty Drupella with 15–17 mm aperture lengths were collected from the fore reef, housed in a glass aquarium with a flow-through sea water system and starved for a minimum of 72 h prior to use in experiments. We placed one P. verrucosa colony from each of the three treatments into the center of three assigned tanks. Three Drupella individuals were lined up on the bottom downstream rear edge of each aquarium to ensure the availability of a chemosensory signal from the coral to the snails. Three Drupella were used in each treatment so that a measurable feeding scar would be produced during the experimental period of 19–24 h. The experiment was repeated each night with a new set of corals and symbionts, with treatments alternating between tanks each day. Drupella were reused after a 72 h starvation period, and a new cohort of Drupella was collected after ten days and starved as above.

After each experiment, the length, width, and height of corals were measured. The size of any feeding scars produced during the experiment was measured. Measurements of the amount of tissue consumed in an irregular three-dimensional branching coral are difficult and contentious. We chose to use proxies of coral volume and feeding scars calculated as the volume of an ellipsoid using the formula: V = 4/3πabc, where a, b, and c are radii following McKeon et al. (2012). After each trial the aquaria were scrubbed, rinsed, drained and refilled.

The defensive efficacy of the crab treatments was compared to appropriately sized control corals; the differences in host size does not allow for direct comparison of the efficacy of small crabs in small corals and larger crabs in larger corals.

Culcita novaeguineae

Experiments testing defense against Culcita by two medium-sized species of Trapezia were conducted in flow-through plastic pools of approximately 2,670 L volume. P. verrucosa were gathered from the back reef environment on the same day of the experiments. Two sets of experiments were conducted with two size classes of corals. Experiments using the larger size-class of coral colonies (mean volume of 3,354 ± 979 cm3) had treatments as follows: pairs of medium-size class T. serenei (n = 20, CW = 9–11 mm), pairs of medium-size class T. bidentata (Forskål, 1775) (n = 17, CW = 9–11 mm), and all symbionts removed (n = 23). The second set of experiments used a smaller size-class of coral (mean volume 519 ± 175 cm3) included pairs of small size-class T. serenei in smaller corals (n = 9, CW = 4–6 mm) and a set with all symbionts removed (n = 8). As above, the desired experimental symbiont community was established through manual removal of symbionts and other coral-associated animals.

Culcita novaeguineae were collected from the back reef (n = 37, mean body diameter 16 cm) in Cook’s Bay, Mo’orea, held in large plastic pools with flowing seawater, and were starved for at least 48 h before experimental use. Corals were positioned in the middle of the plastic pools, and a single Culcita was placed directly on top of each experimental coral at sundown. Predation was evaluated the following morning, after approximately 15 h. Coral volume and feeding scar dimensions were estimated as noted above.

Acanthaster planci

Several studies have suggested that Trapezia may be able to repel Acanthaster corallivory (Glynn, 1987; Pratchett, 2001). These studies have either been observational or conducted as lab experiments. A large outbreak of Acanthaster on Mo’orea beginning in 2008 allowed field assessment of the defensive capabilities of Trapezia flavopunctata Eydoux & Souleyet 1842, the largest species of Trapezia. We chose a spur and groove fore reef site off the northern shore of Mo’orea and conducted experiments from October to November of 2008. We manually removed Trapezia flavopunctata from 45 haphazardly selected Pocillopora eydouxi colonies, and selected a second set of 45 coral colonies with pairs of large size-class T. flavopunctata allowed to remain in the coral. The removals were maintained for one month. Other symbiotic species, including fish, arthropods, and smaller species of Trapezia, were left in the coral colonies. Control corals were disturbed in a manner similar to that used to remove T. flavopunctata. Every 48 h all corals were checked for tissue loss (feeding scars), and measured as described previously.

Results

Drupella cornus

Frequency of predation was significantly higher in corals with symbionts removed (22/22) than in corals containing a small size-class T. serenei pair (8/22; Fisher’s Exact Test, p = 0.0003). Mean tissue loss during predation events was significantly higher in corals with symbionts removed (n = 22, 13.9 cm3, 8.7% total coral volume proxy (TCVP)), than in corals containing small T. serenei pairs (n = 8, 7.94 cm3, 1.3% TCVP; Student’s T-Test p = 0.003, Fig. 1). Neither predation frequency nor mean tissue loss differed significantly between corals containing the medium size-class of T. serenei and the matched set of corals with symbionts removed (predation frequency: Fisher’s Exact Test, p > 0.05; mean tissue loss: Student’s t-test, p > 0.05).

Figure 1 Percentage of coral tissue volume proxy consumed by Drupella cornus in corals hosting the small size-class of Trapezia serenei and Trapezia punctimanus.

Letters indicate post-hoc statistically significant differences between groups.

In trials with corals containing T. punctimanus, predation frequency differed significantly between corals with crab pairs present and removed (Fisher’s Exact Test, p = 0.0009). Mean tissue loss during predation events was also significantly higher in corals with symbionts removed (n = 20, 22.9 cm3, 5.7% of TCVP), than in corals with T. punctimanus pairs present (n = 9, 0.61% of TCVP; Student’s t-test p = 0.00001; Fig. 1).

Culcita novaeguineae

The differences in defense provided to the coral host by the medium size-class of T. serenei and T. bidentata were evaluated using ANOVA. Mean tissue loss differed significantly among treatments: 19% of TCVP in corals containing T. serenei pairs (n = 5), 37% of TCVP in corals containing T. bidentata pairs (n = 10), and 49% of TCVP in corals with symbionts removed (n = 20; ANOVA: p < 0.01, F = 20.77, df = 33; Fig. 2). There was also a significant difference in tissue loss when the data were analyzed as presence or absence of crab pairs, ignoring species differences (Tukey’s HSD, p < 0.001). Smaller corals with the small size-class of T. serenei present (n = 9) and removal treatments (n = 8) were completely consumed by Culcita, so no statistical test was needed.

Figure 2 Percentage of coral tissue volume proxy consumed by Culcita novaeguineae in corals hosting the large size-class of Trapezia bidentata and Trapezia serenei.

Letters indicate post-hoc statistically significant differences between groups.

Acanthaster planci

Live coral cover on the fore reef of Mo’orea from 2000 to 2006 varied from 41 to 51%, and was dominated by species of Pocillopora, Porites, and Acropora (Adjeroud et al., 2009). Photos taken in 2006 of our immediate study area provided estimated coral cover of about 80%. By September 2008, coral cover had plummeted in northern Mo’orea as a result of a population outbreak of Acanthaster planci. In our study area, total living coral cover had decreased to 3.4% (se = 0.97, n = 45) as calculated from quadrat surveys, and most of the coral recorded during the survey were P. eydouxi. Living coral cover changed little by October 2009; with estimated cover at 3.2% (se = 0.75, n = 45), and Pocillopora eydouxi remained the most abundant coral species on the reef.

Removal of T. flavopunctata from remaining live P. eydouxi led to an increased rate of attack and tissue loss in hosts. Over the two-week experimental period, 64% (29/45) of corals with symbionts removed were attacked, compared with 18% (8/45) of corals with T. flavopunctata pairs present (Binomial Proportions Test, χ2 = 18.358, df = 1, p < 0.005; Supplemental Information 1). Mean coral tissue loss was 22% of TCVP in undefended corals and 2% of TCVP in defended corals (Binomial Proportions Test, p < 0.005; Supplemental Information 2). Thirty-four corals from which T. flavopunctata were removed still possessed a complement of other Trapezia species and other symbiotic taxa. We compared attack frequency between P. eydouxi with T. flavopunctata and smaller-size Trapezia species present (10 attacks/33 corals), to corals from which T. flavopunctata had been removed, but with other symbionts left alone (24 attacks/34 corals). Corals with T. flavopunctata removed suffered attack by Acanthaster more frequently (Binomial Proportions Test, χ2 = 9.3214, df = 1, p = 0.0022).

Discussion

Efficacy of defense against corallivores differed among Trapezia species as well as among size classes. Small (CW = 4–6 mm) T. punctimanus and T. serenei were comparably effective in defending against Drupella, while larger (CW = 9–10 mm) T. serenei were ineffective. However, large (CW = 9–11 mm) T. serenei were effective in defending against Culcita, while small T. serenei (CW = 4–6 mm) were not. Furthermore, large (CW = 9–11 mm) T. serenei were significantly more effective host defenders than comparably sized T. bidentata against Culcita predation, suggesting that specific identity, as well as size plays a role in the level of protective service provided to the coral host. Finally, during the Acanthaster outbreak only corals hosting the largest Trapezia species, i.e., T. flavopunctata or T. rufopunctata, survived. Smaller-size Trapezia species were unable to protect their hosts against the largest of the three predators, and the corals were quickly attacked when T. flavopunctata were removed.

These results suggest ecological complementarity, as well as a hierarchy of defensive effectiveness among different species and sizes of Trapezia. Small crabs effectively defend their hosts against the small predator Drupella, but fail against larger predators such as Culcita: small Pocillopora were always consumed entirely by Culcita, regardless of the presence or absence of small Trapezia. The crabs flee the coral, or are consumed along with the host.

Medium sized crabs are effective against Culcita, with effectiveness varying among species, but they do not defend against Drupella or Acanthaster. The largest crabs actively defend against Acanthaster, but their efficacy against the other coral predators remains untested. Thus, barring the untested possibility of negative interactions, a coral harboring all of these symbionts may be defended against all three corallivores, while a coral with a lesser complement of Trapezia species may remain vulnerable to some predators. An additional aspect of this within guild complementarity, or ‘species stacking’, is that several species may create additional synergistic defensive effects even against the same coral predator (McKeon et al., 2012).

These results also suggest that the characteristics of the mutualism between Trapezia and Pocillopora may shift as the resident crabs and coral hosts increase in size. Because the feeding scars of the corals without symbionts in the small T. serenei and T. punctimanus experiments were nearly the same size as the feeding scars in both groups of the large T. serenei experiments, it may be possible that the threat to smaller crabs and their correspondingly smaller hosts by Drupella predation is proportionally greater. This proportional response may explain why larger crabs did not respond as effectively as the smaller crabs to Drupella. Glynn (1980) suggested that another small species of Trapezia, T. formosa may not play a role in defending their host corals against Acanthaster. However, we have shown that smaller species of Trapezia previously thought to be ineffective guards, because they were only tested against Acanthaster, can play important defensive roles against other corallivores.

The efficacy of protection provided by two species of Trapezia (T. serenei and T. bidentata) against Culcita differed under controlled conditions. Both significantly reduced the frequency of attack and volume of tissue consumed by Culcita, but T. serenei was more effective. This may be because T. serenei is more common in the back reef habitat where Culcita populations are most dense, while T. bidentata is more common in fore reef environments where Culcita is less frequently encountered in Mo’orea.

Both attack rate and the amount of coral tissue consumed by Acanthaster were significantly reduced by the presence of the largest species in the system, Trapezia flavopunctata. Acanthaster preyed upon P. eydouxi when Trapezia flavopunctata were removed. Species of Pocillopora that do not host T. flavopunctata were completely consumed by Acanthaster in the reef area studied, despite the frequency of occupancy of other Trapezia spp. exceeding 90% (Stewart et al., 2006; McKeon et al., 2012).

The protective impact of T. flavopunctata appeared to extend beyond the host coral, providing protection for a microcommunity of reef corals in the vicinity of their host. In a survey conducted at the same time as these field trials, 61 colonies of 13 other reef coral species, eliminated from the reef in all other areas by Acanthaster, were found under and immediately beside of 90 P. eydouxi-symbiont communities (Fig. 3). No living corals were found in similar positions around 90 recently dead P. eydouxi of similar size in the same area. Indirect, associational defense has previously been documented among reef corals in the same location (Kayal et al., 2010). Protection provided by the presence of the largest species of Trapezia is another potential mechanism for the survival of reef corals in Acanthaster outbreaks.

Figure 3 Examples of corals sheltered by living Pocillopora eydouxi and Trapezia flavopunctata.

The results of these experiments raise many questions about the Trapezia–Pocillopora mutualism. Glynn (1980) demonstrated that Trapezia respond aggressively to chemical signals produced by Acanthaster, but whether Trapezia are able to detect and respond to chemical signals produced by other corallivores such as Drupella remains unknown.

During an analysis of coral population dynamics in Australia, Hughes & Connell (1987) found that 39% of Pocillopora damicornis measuring less than 10 cm2 died over the course of one year, while only 8% of P. damicornis measuring between 10 and 50 cm2 suffered the same fate. While differential mortality is expected with changes in colony size, impacts of mutualists on different sizes of corals should be examined. The frequency of occurrence of Trapezia in medium-sized Pocillopora colonies is well documented (Huber & Coles, 1985; Sin & Lee, 2000; Stewart et al., 2006) but our understanding of the occurrence of Trapezia in very small corals is limited. In recent work by Stewart et al. (2013), growth of small Pocillopora was positively impacted by the presence of recently recruited Trapezia serenei through the removal of sediment. Similar work on the ontogeny of defense from corallivores is needed.

While the largest crab species provide the greatest defensive efficacy against Acanthaster, the energy expenditure of the host coral to maintain the symbiosis may also be costly. As the frequency of Acanthaster outbreaks on Mo’orea has been on an approximately 30-year cycle (Faurea, 1989) attacks by seastars may be relatively rare events in the life of a coral. Removal of sediment and deterrence of smaller corallivores, such as Drupella, may be more common needs for coral, even if the size of their impact on a colony may not be as severe as that resulting from Acanthaster attack. The beneficial role of different symbiont species and size classes to this spectrum of impact may be sufficient to increase the fitness of corals that maintain the nutritive benefits that are provided to the crabs (Stimson, 1990).

The experiments we conducted suggest that functional diversity exists in the Pocillopora–Trapezia mutualism at a species level. But we must acknowledge the limitations of our study: we were unable to use all species of Trapezia present in the system, in a single size and species of Pocillopora. Variation within the mutualism was demonstrated to impact coral host survival across the size ranges of host and symbiont. Interactions between the species and size classes are likely an integral part of the mechanisms that promote the existence of the Pocillopora–Trapezia mutualism, but have yet to be explored in depth. Variation in the services provided by mutualists has been recorded in a wide variety of systems (Addicott, 1986; Bronstein & Hossaert-McKey, 1996; Del-Claro & Oliveira, 2000). Therefore, the assumptions underlying construction of functional groups may be weaker than generally noted in the literature, and functional groups of questionable utility in understanding diverse mutualist systems.

Our studies suggest that measuring a diversity of response variables is important in assessing the level of functional diversity present in a system. Our results demonstrate the importance of the Trapezia–Pocillopora mutualism in response to corallivory, including an outbreak of Acanthaster. A diverse symbiont fauna may provide individual corals with greater protection from a suite of predators. Moreover, these benefits may cascade beyond the individual coral colony and have far-reaching impacts on the reef, by altering the demography of small corals, structuring the surviving communities of corals and offering associational defenses and refuge to nearby corals during catastrophic predator outbreaks.

Supplemental Information

Supplemental Information 1 Defense of Pocillopora eydouxi by Trapezia flavopunctata

Click here for additional data file.

Supplemental Information 2 Data set of T. flavopunctata removal experiment

Click here for additional data file.

We are grateful to Gustav Paulay, François Michonneau, Mike Gil, and Nat Seavy for their suggestions and comments on the manuscript. We would also like to thank the staff of UC Berkeley’s Richard B. Gump South Pacific Research Station.

Additional Information and Declarations

Competing Interests

Author Contributions

Field Study Permissions

The authors declare there are no competing interests.

C. Seabird McKeon and Jenna M. Moore conceived and designed the experiments, performed the experiments, analyzed the data, contributed reagents/materials/analysis tools, wrote the paper, prepared figures and/or tables, reviewed drafts of the paper.

The following information was supplied relating to field study approvals (i.e., approving body and any reference numbers):

Haut-commissariat de la République en Polynésie française,

Le délégué régional à la recherche et à la technologie de la Polynésie française,

Protocole D’accueil D’un Chercheur ou Enseignant-Chercheur Etranger.

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
