# Peer review of "Species and size diversity in protective services offered by coral guard-crabs"

_PeerJ, doi:10.7717/peerj.574_

## Round 0.1 · original submission · Minor Revisions

Please pay careful attention to the description of the experimental design and where appropriate any interpretation in the discussion as outlined by reviewer 2.

·

Basic reporting

There are some issues of varying importance that need to be solved:
1. line 4 and elsewhere: “coral-guard crabs” OR “coral guard-crabs” NOT “coral guard crabs” – subtle but important difference!
2. line 9 and elsewhere: “Mo‘orea” not “Mo’orea” – use of okina ["inverted comma"] not apostrophe
3. line 12 and elsewhere: “smaller Trapezia species,” “small Trapezia species” – use “smaller-size...” and “small-size ...”
4. line 24: “arms races” – delete term (anthropomorphism) or, at least, use quotation marks for the term
5. line 60 – give reference(s)for figures on number of Trapezia spp.: there are 22 speciesw, 14 of which have been recorded from French Polynesia (high number of endemics, perhaps the result of peripheral endemicity
6. line 83 and elsewhere: why “Central Pacific”? For one thing French Polynesia is not in the Central Pacific, secondly, the distribution of Pocillopora you describe can be applied to most in not all shallow-water reefs in the Indo-West Pacific (and Tropical Eastern Pacific) regions
7. line 109: I wouldn’t call T. punctimanus a “small[-size] species”. Size is relative but the maximum adult size of species like T. formosa and T. globosa is much, much smaller than that of T. punctimanus and the congeners
8.line 352: a lapsus must have occurred when you, out of nowhere, refer to a “commensal fauna”! Musualistic or symbiotic fauna but NOT commensal, please! This is an old-fashioned, catch-all term that should not be used in this context.

Experimental design

It looks fine to me. I wish the authors had used the truly small-size species such as T. formosa and T. globosa, or even T. guttata. The first and last species are more common on live coral fragments.

Validity of the findings

No comments. In any case, most important and interesting results.

Additional comments

Excellent piece of work. I wish you had used the truly small-size species such as T. formosa and T. globosa, or even T. guttata. The first and last species are more common on live coral fragments. I wish you had speculated, by giving a few hypotheses, why different species of Trapezia of approximately the same size (and supposedly similar food habits) react differently to coral predators. For further study (though a difficult task indeed): analyze the energetics of the association: do all species and all size categories of Trapezia have the same metabolism and therefore the same energy requirements? I doubt so, but the question is why? Do all ingest fat bodies? Are there any food specialists among the species? Do different parts of the coral colony produce less mucus than others? Do some species of coral produce more coral mucus or fat bodies?

Reviewer 2 ·

Basic reporting

This manuscript contains a nice overview to the current state of knowledge on coral-crab mutualisms, and the authors clearly know their subject (some recent work on small crab- juvenile coral associations published in MEPS recently could be added ~ lines 326-328).

See experimental design.

Some small edits:
Some text needs to be clarified. On line 112, CW appears without having been previously defined. Also, line 142 refers to methods ‘as described above,’ but they are not.

Line 159 states that the number of snails found on the corals was recorded, but these numbers are not reported.

Experimental design

This is a complicated study, and as the authors acknowledge, they were unable to design a fully factorial design to address their goals. This is a function of a natural system, and is not surprising. Given the complexity of their work, though, the authors have to take extra steps to be very clear about their experimental design. I found it confusing, having to read and re-read, and re-read again the methods to understand which size of which crab species was used with which size coral for each predator. I ended up drawing out each experiment, and again this took a lot of effort, as it is presented in a confusing manner in the text. For example, crabs are initially introduced by species in size groupings (small, medium, large)(lines 109-114), but then these groupings aren’t necessarily retained within descriptions of individual experiments. For example, crabs were described as the ‘larger’ size class within an experiment, when that experiment was said to be using the medium and small sized crabs (Drupella experiment, lines 143-147). In the Culcita experiment, crab carapace widths are given for the crabs used without reference to their original categories of small, medium and large, and the original designations do not include carapace width size ranges. These are examples of what makes following the experimental design of this manuscript very confusing. It could be a relatively simple fix, though, by clear initial descriptions followed by consistency in referring to the experimental groups. A diagram of the players and the experimental design for each experiment would also be valuable. It’s clear that the authors know a lot about the taxonomy and natural history of these species, and this is part of the point that they are making about species functional diversity/redundancy, but they need to work to find ways to confine the details within tractable units.

Validity of the findings

This is an ambitious project and contains important information about coral-crab mutualisms. The finding that smaller crab species can play defensive roles against reef predators other than Acanthaster is valid and important. The ecological relevance of this work could be strengthened by acknowledging the relevance of their findings to the well-known survival bottleneck for coral recruits at small/young stages. They refer to differential survival with age of corals on lines 320-328, and couching their findings within this theme would serve them well.

Relating the efficacy of host protection results to the various distributions of crabs and Culcita on the reef is a nice natural history tie-in that further validates the relevance of findings. This is well done in many parts of the discussion.

The data collected in the forereef surveys do not show associational defence. While these surveys do show that smaller corals are present within the vicinity of live corals, and their experiments do show that corals with crabs survive better than those without, they do not show causation or evidence of crab protective ability as the mechanism responsible for this pattern, reported and tested experimentally in Kayal et al. 2010. It is possible that defence by associate species is a factor responsible for this pattern, and this should absolutely be included in the discussion as speculation, but removed as a finding (and objective) of this study.

Additional comments

This is a novel approach to examining coral-crab mutualisms and is important for larger ecological understanding of the role and resilience of services provided through mutualistic associations. It’s a natural, (i.e. messy) system that presents many challenges, but this type of work is necessary to address the complexity of this system.

The authors need to be careful not to overstep the limitations of their data in their conclusions (as noted above re: forereef survey). Overall this is an ambitious project that attempts to determine functional diversity of a well-known mutualism. While these data certainly contribute to this larger theme, they are not, in themselves enough to confirm or refute functional redundancy/diversity, but are a valuable contribution

---

## Round 0.2 · Minor Revisions

Dear Authors,

Thank you for returning your revised manuscript and addressing all the comments of the reviewers. I would kindly ask whether you, prior to acceptance, you could include the test statistics (e.g. F= ?, df = ?, p < 0.01) for your ANOVAs and Chi-Square analyses in your manuscript either in tabulated form or within the text .

---

## Round 0.3 · accepted · Accept

Dear Authors,

Thank you on making those additional corrections to the manuscript.